# Clinical characteristics and treatment outcomes of *Acinetobacter baumannii* bloodstream infections in a setting with high carbapenem susceptibility among isolates

Jinghao Nicholas Ngiam,[1] Matthew Chung Yi Koh,[1] Ka Lip Chew[2]

**ABSTRACT** In most settings, carbapenem-resistant *Acinetobacter baumannii* (CRAB) infections predominate over carbapenem-susceptible *Acinetobacter baumannii* (CSAB). Treatment guidelines focus on the management of CRAB and do not describe the optimal antibiotic choice for CSAB. We describe the clinical characteristics and outcomes in both CRAB and CSAB. We examined consecutive episodes of CRAB or CSAB bloodstream infections in our institution from February 2022 to July 2024. Clinical, laboratory, and microbiological data were tabulated, and adverse outcomes were defined as the need for intensive care (ICU) or all-cause in-hospital mortality. We compared (i) CRAB to CSAB, and among patients with CSAB, (ii) we evaluated if the definitive antibiotic choice was associated with adverse outcomes, after adjusting for C-reactive protein levels. We identified 71 unique episodes of bacteremia, of which a majority (52/71, 73.2%) were nosocomial onset. A minority were CRAB (15/71, 21.1%). These patients often had ICU onset, prior carbapenem exposure, and were more likely to experience mortality (73.3% versus 16.1%, *P* < 0.001). Only a minority of patients (7/71, 9.9%) received dual antibiotic therapy. Among the 56 patients with CSAB, only 11/56 (19.6%) experienced adverse outcomes. On multivariable analysis, after adjusting for elevated C-reactive protein, the use of carbapenems as the definitive antibiotic choice remained independently associated with adverse outcomes (adjusted odds ratio 7.34, 95% CI 1.25–43.01, *P* = 0.027). CSAB was far more common than CRAB in our setting. Prior carbapenem exposure and ICU onset were important risk factors for CRAB, which had higher mortality than CSAB. Carbapenem-sparing options may be considered as the definitive antibiotic choice for CSAB.

**IMPORTANCE** In our setting, carbapenem-susceptible *Acinetobacter baumannii* (CSAB, *n* = 56/71, 78.9%) bloodstream infections (BSIs) were far more common than with carbapenem-resistant isolates (CRAB). CRAB BSI was associated with intensive care onset, prior carbapenem use, and had higher mortality (73.3% versus 16.1%). In CSAB BSI, after adjusting for elevated C-reactive protein, the definitive use of carbapenems for treatment remained independently associated with adverse outcomes (adjusted odds ratio 7.34, 95%CI 1.25–43.01). These findings may suggest that carbapenem-sparing regimens may be effective alternative antibiotic choices for CSAB. Although treatment guidelines focus on CRAB, future prospective studies should also evaluate optimal treatment strategies for CSAB.

**KEYWORDS** carbapenem-resistant, carbapenem-sensitive, *Acinetobacter baumannii*, outcomes, antibiotic choice

Carbapenem-resistant *Acinetobacter baumannii* (CRAB) remains a major cause of healthcare-associated infections, particularly in critically ill patients. In most clinical

**Peer Reviewers** Maurizio Sanguinetti, Fondazione Policlinico Universitario Agostino Gemelli IRCCS, Rome, Italy; Israa M.S. AL-Kadmy, Plymouth University, Plymouth, United Kingdom

Address correspondence to Jinghao Nicholas Ngiam, nicholas_ngiam@nuhs.edu.sg, or Matthew Chung Yi Koh, matthew.koh@mohh.com.sg.

Jinghao Nicholas Ngiam and Matthew Chung Yi Koh contributed equally to this article. The author order was determined by seniority.

The authors declare no conflict of interest.

settings, CRAB may account for over 80% of clinical isolates, compared with carbapenem-susceptible *Acinetobacter baumannii* (CSAB) (1–3). CRAB has significantly higher mortality and adverse outcomes compared with CSAB (2, 4). Consequently, guidelines from international societies focus on the management of CRAB, often recommending combination antimicrobial therapy, involving sulbactam in combination with another active agent (5, 6).

In comparison, the optimal antibiotic choice and clinical outcomes in CSAB are ill-defined, as they often form the minority of patients in cohorts. Uniquely, in our single-center experience, CSAB accounted for more than half of the identified clinical isolates of *Acinetobacter baumannii* (7). As such, we were well positioned to compare the clinical features of CRAB versus CSAB. Additionally, within patients with CSAB bloodstream infections, we sought to define parameters that were associated with adverse outcomes.

We retrospectively examined consecutive cases of *Acinetobacter baumannii* bloodstream infections. Blood culture isolates were identified using MALDI-TOF mass spectrometry (Bruker MALDI Biotyper, Bruker, Billerica, MA, USA). Routine susceptibility testing was performed with VITEK II (bioMérieux, Marcy-l'Étoile, France) and interpreted according to European Committee on Antimicrobial Susceptibility Testing breakpoints where breakpoints were available (8). For ceftazidime and piperacillin-tazobactam, isolates were considered resistant if the MICs were higher than the epidemiological cut-off value. All patients were managed at a single tertiary care hospital from February 2022 to July 2024 and were adults over 21 years of age. Clinical data were obtained from a retrospective review of the medical records. All patients were managed according to institutional infection prevention measures, where patients with CRAB were put on contact precautions, with single-room isolation where feasible. The choice of definitive antibiotic therapy was selected by the managing physicians, guided by *in vitro* susceptibility, clinical status, and any drug allergies. Data on the background, infection source, antimicrobial treatment strategy, and duration were tabulated. Adverse clinical outcomes were defined as all-cause in-hospital mortality or the need for intensive care.

For analysis, the cohort was first divided into CRAB (defined by resistance to meropenem) and CSAB. Further subgroup analysis was performed on the subgroup of patients with CSAB, dividing them into those who experienced adverse outcomes (in-hospital mortality or required intensive care) and those who did not. For the pairwise comparisons, continuous parameters were evaluated with *t*-tests, and categorical parameters were evaluated with Chi-squared tests. To determine if definitive antibiotic choice was associated with adverse outcomes in CSAB, we constructed a multivariable model in the CSAB subgroup, including only C-reactive protein as a covariate due to the small sample size and low rate of adverse outcomes, in order to minimize the risk of model overfitting. All analysis was conducted on SPSS version 20.0. A *P*-value of <0.05 was considered significant.

We identified 71 unique episodes of bloodstream infections with *Acinetobacter baumannii*. Of these 15/71 (21.1%) were CRAB, with the majority of the isolates being CSAB. There were no significant differences in age, sex, or comorbidities. Compared to CSAB, risk factors for CRAB bacteremia include ICU onset (46.7% versus 12.5%, *P* = 0.003), prior carbapenem exposure (86.7% versus 17.9%, *P* < 0.001), or the presence of a central venous catheter (40.0% versus 1.8%, *P* < 0.001). Mortality rates were significantly higher in CRAB (73.3% versus 16.1%, *P* < 0.001) (Table 1). Initial dual antibiotic therapy was far more common in CRAB than in CSAB (40.0% versus 1.8%). This difference reflects recommendations from treatment guidelines for the management of CRAB, while no such guidelines exist for CSAB (5, 6).

Definitive treatment of CRAB and CSAB varied. Of the patients with CRAB bacteremia, 40% (6/15) and 33.3% (5/15) received ampicillin-sulbactam and polymyxin-based therapy, respectively. Comparatively, patients with CSAB bacteremia were less likely to receive ampicillin-sulbactam (3/56, 5.4%) and polymyxin-based therapy (0/56, 0.0%), and instead received carbapenem (7/56, 12.5%), cephalosporins (ceftazidime or cefepime,

**TABLE 1** Clinical characteristics of carbapenem-resistant *Acinetobacter baumannii* versus carbapenem-sensitive *Acinetobacter baumannii* bloodstream infections

| Parameter | Overall (*n* = 71) | CRAB (*n* = 15) | CSAB (*n* = 56) | *P*-value |
|---|---|---|---|---|
| Clinical presentation | | | | |
| Age (years) | 66.7 (±15.1) | 66.8 (±9.2) | 66.7 (±16.4) | 0.985 |
| Male sex | 42 (59.2%) | 8 (53.3%) | 34 (60.7%) | 0.606 |
| Hypertension | 32 (45.7%) | 9 (60.0%) | 23 (41.8%) | 0.210 |
| Hyperlipidemia | 36 (50.7%) | 7 (46.7%) | 29 (51.8%) | 0.725 |
| Diabetes mellitus | 30 (42.3%) | 9 (60.0%) | 21 (37.5%) | 0.117 |
| End-stage kidney disease | 12 (16.9%) | 3 (20.0%) | 9 (16.1%) | 0.718 |
| Ischemic heart disease | 22 (31.0%) | 4 (26.7%) | 18 (32.1%) | 0.684 |
| Immunocompromised host | 36 (50.7%) | 9 (60.0%) | 27 (48.2%) | 0.417 |
| Intensive care onset | 14 (19.7%) | 7 (46.7%) | 7 (12.5%) | 0.003 |
| Nosocomial onset | 52 (73.2%) | 13 (86.7%) | 39 (69.6%) | 0.186 |
| Presence of a central line | 7 (9.9%) | 6 (40.0%) | 1 (1.8%) | <0.001 |
| Source of infection | | | | 0.247 |
| Unclear | 0 (2.8%) | 0 (0.0%) | 2 (3.6%) | |
| Central line | 22 (31.0%) | 2 (13.3%) | 20 (35.7%) | |
| Pneumonia | 13 (18.3%) | 5 (33.3%) | 8 (14.3%) | |
| Skin and soft tissue | 14 (19.7%) | 2 (13.3%) | 12 (21.4%) | |
| Intra-abdominal | 18 (25.4%) | 5 (33.3%) | 13 (23.2%) | |
| Urinary | 2 (2.8%) | 1 (6.7%) | 1 (1.8%) | |
| Prior carbapenem exposure in the last 1 month | 23 (32.4%) | 13 (86.7%) | 10 (17.9%) | <0.001 |
| Temperature (°C) | 38.1 (±0.8) | 38.0 (±1.0) | 38.2 (±0.8) | 0.525 |
| Systolic blood pressure (mmHg) | 123 (±26) | 110 (±25) | 126 (±25) | 0.029 |
| Pulse rate (per minute) | 90 (±21) | 98 (±23) | 88 (±20) | 0.111 |
| Total white cell count (×10$^9$/L) | 10.3 (±6.9) | 8.7 (±7.4) | 10.7 (±6.7) | 0.328 |
| Serum creatinine (µmol/L) | 150 (±185) | 146 (±106) | 151 (±201) | 0.923 |
| C-reactive protein (mg/L) | 87.7 (±93.4) | 141.5 (±111.0) | 73.2 (±83.4) | 0.011 |
| Microbiology | | | | |
| Polymicrobial bacteremia | 19 (26.8%) | 3 (20.0%) | 16 (28.6%) | 0.505 |
| Resistant to cefepime | 19 (26.8%) | 15 (100.0%) | 4 (7.1%) | <0.001 |
| Resistant to ceftazidime | 28 (39.4%) | 15 (100.0%) | 13 (23.2%) | <0.001 |
| Resistant to ceftazidime-avibactam | –[a] | 11 (73.3%) | – | – |
| Resistant to ceftolozane-tazobactam | – | 11 (73.3%) | – | – |
| Resistant to ciprofloxacin | 16 (22.5%) | 15 (100.0%) | 1 (1.8%) | <0.001 |
| Resistant to meropenem | 15 (21.1%) | 15 (100.0%) | 0 (0.0%) | <0.001 |
| Resistant to piperacillin-tazobactam | 25 (35.2%) | 15 (100.0%) | 10 (17.9%) | <0.001 |
| Resistant to tigecycline | – | 2/15 (13.3%) | – | – |
| Resistant to trimethoprim-sulfamethoxazole | 15 (21.1%) | 10 (66.7%) | 5 (8.9%) | <0.001 |
| Treatment and outcomes | | | | |
| Total antibiotic duration (days) | 11.1 (±6.6) | 7.8 (±3.7) | 12.0 (±7.0) | 0.028 |
| Received dual antibiotic therapy | 7 (9.9%) | 6 (40.0%) | 1 (1.8%) | <0.001 |
| Received intervention for source control of infection (e.g., line removal and drainage of abscess) | 42 (59.2%) | 4 (26.7%) | 38 (67.9%) | 0.004 |
| Required intensive care | 17 (23.9%) | 7 (46.7%) | 10 (17.9%) | 0.020 |
| All-cause in-hospital mortality | 20 (28.2%) | 11 (73.3%) | 9 (16.1%) | <0.001 |

[a]–, not applicable/not tested.

13/56, 23.2%), piperacillin-tazobactam (4/56, 7.1%), ciprofloxacin (23/56, 41.1%), and other antibiotics. Most CSAB isolates remained susceptible to ceftazidime (76.8%), piperacillin-tazobactam (82.1%), and ciprofloxacin (98.2%) (Table 1).

 Among patients with CSAB bacteremia, 11/56 (19.6%) experienced adverse outcomes (requiring intensive care or all-cause in-hospital mortality). Adverse outcomes were more

likely in patients with elevated inflammatory markers at the onset of bacteremia, such as total white cell count (15.3 ± 9.9 versus ±9.6 ± 5.3 × $10^9$/L, $P$ = 0.012) and C-reactive protein (129.8 ± 134.2 versus 59.4 ± 60.0 mg/L, $P$ = 0.011). Definitive antibiotic choice with a carbapenem was also associated with higher mortality (36.4% versus 6.7%, $P$ = 0.008). Only one patient received dual antibiotic therapy for CSAB (Table 2). On multivariable analyses, after adjusting for C-reactive protein levels, definitive antibiotic choice with carbapenems remained independently associated with adverse outcomes (adjusted odds ratio 7.34, 95% CI 1.25–43.01, $P$ = 0.027).

Our findings highlight the significant clinical implications of *Acinetobacter baumannii* infections, particularly the differences in outcomes between CRAB and CSAB. We uniquely describe a clinical setting where CSAB predominates and forms ~80% of our cohort of bloodstream infections, over CRAB. As expected, prior carbapenem exposure was an important predisposition for CRAB over CSAB. Additionally, CRAB was also more common in the ICU, and from a prior study, shown to be less genetically diverse. This highlights the importance of antibiotic stewardship, as well as infection control measures to limit the spread and selection for this organism (7, 9, 10). The increased mortality we observed with CRAB compared with CSAB is also in line with prior work (4).

**TABLE 2** Clinical characteristics of *Acinetobacter baumannii* bloodstream infections by adverse outcomes (all-cause in-hospital mortality or requiring intensive care)

| Parameter | Adverse outcomes ($n$ = 11) | No adverse outcomes ($n$ = 45) | $P$-value |
|---|---|---|---|
| Age (years) | 69.8 (±14.7) | 66.0 (±16.9) | 0.490 |
| Male sex | 6 (54.5%) | 28 (62.2%) | 0.640 |
| Hypertension | 4 (40.0%) | 19 (42.2%) | 0.897 |
| Hyperlipidemia | 5 (45.5%) | 24 (53.3%) | 0.639 |
| Diabetes mellitus | 5 (45.5%) | 16 (35.6%) | 0.543 |
| End-stage kidney disease | 1 (9.1%) | 8 (17.8%) | 0.482 |
| Ischemic heart disease | 6 (54.5%) | 12 (26.7%) | 0.076 |
| Immunocompromised host | 2 (18.2%) | 25 (55.6%) | 0.026 |
| Nosocomial onset | 3 (27.3%) | 14 (31.1%) | 0.804 |
| Presence of a central line | 6 (54.5%) | 19 (42.2%) | 0.461 |
| Source of infection | | | 0.240 |
| Unclear | 0 (0.0%) | 2 (4.4%) | |
| Central line | 3 (27.3%) | 17 (37.8%) | |
| Pneumonia | 4 (36.4%) | 4 (8.9%) | |
| Skin and soft tissue | 1 (9.1%) | 11 (24.4%) | |
| Intra-abdominal | 3 (27.3%) | 10 (22.2%) | |
| Urinary | 0 (0.0%) | 1 (2.2%) | |
| Polymicrobial bacteremia | 4 (36.4%) | 12 (26.7%) | 0.523 |
| Temperature (°C) | 38.3 (±0.5) | 38.1 (±0.9) | 0.387 |
| Systolic blood pressure (mmHg) | 111 (±29) | 130 (±22) | 0.024 |
| Pulse rate (per minute) | 98 (±22) | 85 (±19) | 0.051 |
| Total white cell count (×$10^9$/L) | 15.3 (±9.9) | 9.6 (±5.3) | 0.012 |
| Serum creatinine (µmol/L) | 119 (±72) | 159 (±222) | 0.565 |
| C-reactive protein (mg/L) | 129.8 (±134.2) | 59.4 (±60.0) | 0.011 |
| Total antibiotic duration (days) | 13.8 (±9.5) | 11.6 (±6.3) | 0.345 |
| Received carbapenem as empirical antibiotic choice | 7 (63.6%) | 29 (64.4%) | 0.960 |
| Received carbapenem as definitive antibiotic choice | 4 (36.4%) | 3 (6.7%) | 0.008 |
| Received dual antibiotics | 1 (9.1%) | 0 (0.0%) | 0.196 |
| Required intervention for source control of infection (e.g., line removal and drainage of abscess) | 4 (36.4%) | 34 (75.6%) | 0.013 |

The phenomenon where CSAB predominates over CRAB is rarely reported in the rest of the world (1–3, 11). Because CSAB is rarely described, the optimal antibiotic choice for CSAB remains unclear. Some previous studies suggest carbapenems as first line, with ampicillin-sulbactam being an alternative (12, 13). *In vitro* studies also support the use of third-generation cephalosporins like ceftazidime, although clinical data to support this remain lacking (14). Synergistic use of dual antibiotic therapy, for example, with minocycline and polymyxin B or meropenem or sulbactam, has also been demonstrated *in vitro*, but similarly, its routine use is of uncertain clinical benefit (15). In our small observational cohort, we observed that definitive use of carbapenems was associated with higher mortality, compared with non-carbapenem antibiotics, even after adjusting for C-reactive protein levels, which would be a marker of infection severity. The high rates of susceptibility to ceftazidime, piperacillin-tazobactam, and ciprofloxacin among CSAB in our cohort support the clinical use of these agents as effective carbapenem-sparing options. The majority of patients with CSAB bacteremia also received monotherapy with overall positive outcomes.

While appropriate and effective treatment of infections is important, our data also highlight that prevention of infections with CRAB is also important. While we did not perform genotypic characterization of the isolates in this study, recent local molecular epidemiology data from our institution have demonstrated that the majority of CRAB isolates are OXA-23 carbapenemase producers with limited genetic diversity, whereas CSAB isolates are more heterogeneous in genotype and resistance phenotype (7). Incorporating genomic analyses in future work would allow a better understanding of the resistance mechanisms and transmission dynamics in CRAB.

Furthermore, our current study is limited by the relatively small sample size, particularly the CRAB group ($n = 15$), and a single-center, retrospective design limits statistical power and the generalizability of our findings. Although we adjusted for C-reactive protein as a marker of severity, patients receiving carbapenems as definitive therapy for CSAB may have been more severely ill, introducing confounding by indication. We did not systematically capture other severity indices such as SOFA or APACHE II scores, or the time to appropriate antibiotics, as some of the information was not readily available from retrospective review of the electronic medical records. As such, causality could not be clearly determined. A future prospective study is needed to validate our findings and clearly define optimal antibiotic therapy for this infection.

In summary, we report a cohort of *Acinetobacter baumannii* bacteremia with a high proportion of CSAB (~80%). Prior carbapenem exposure and ICU onset were associated with CRAB, which had higher mortality. Comparatively, mortality was lower in CSAB bacteremia, for which most patients received non-carbapenem monotherapy. Carbapenem-sparing agents may be used for the treatment of CSAB.

## ACKNOWLEDGMENTS

M.C.Y.K., J.N.N., and K.L.C contributed to the conception, data collection, data analysis, original draft, and critical review of the manuscript.

## AUTHOR AFFILIATIONS

[1]Division of Infectious Diseases, Department of Medicine, National University Health System, Singapore, Singapore
[2]Department of Laboratory Medicine, National University Hospital, Singapore, Singapore

## AUTHOR ORCIDs

Jinghao Nicholas Ngiam http://orcid.org/0000-0002-3339-7281
Matthew Chung Yi Koh http://orcid.org/0009-0002-7374-8207

## AUTHOR CONTRIBUTIONS

Jinghao Nicholas Ngiam, Conceptualization, Data curation, Formal analysis, Writing – original draft, Writing – review and editing | Matthew Chung Yi Koh, Conceptualization, Data curation, Formal analysis, Writing – original draft, Writing – review and editing | Ka Lip Chew, Conceptualization, Data curation, Formal analysis, Writing – review and editing

## DATA AVAILABILITY

Data from this study may be made available from the corresponding author upon reasonable request.

## ETHICS APPROVAL

This study was approved by the hospital's institutional review board (National Healthcare Group (NHG) Domain Specific Review Board (DSRB) reference number 2024-3788. A waiver of informed consent was obtained from the above review board prior to the conduct of this retrospective study.

## ADDITIONAL FILES

The following material is available online.

### Open Peer Review

**PEER REVIEW HISTORY (review-history.pdf).** An accounting of the reviewer comments and feedback.

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
