## [Reviewer comments · Microbiology Spectrum]

Microbiology Spectrum

Clinical characteristics and treatment outcomes of *Acinetobacter baumannii* bloodstream infections in a setting with high carbapenem-susceptibility among isolates

Jinghao Nicholas Ngiam, Matthew Koh, and Ka Chew

Corresponding Author(s): Jinghao Nicholas Ngiam, National University Health System

Review Timeline:

Submission Date:	May 23, 2025
Editorial Decision:	August 11, 2025
Revision Received:	August 18, 2025
Accepted:	September 6, 2025

Editor: Po-Yu Liu

Reviewer(s): Disclosure of reviewer identity is with reference to reviewer comments included in decision letter(s). The following individuals involved in review of your submission have agreed to reveal their identity: Maurizio Sanguinetti (Reviewer #1); Israa M.S. AL-Kadmy (Reviewer #2)

Transaction Report:

DOI: <https://doi.org/10.1128/spectrum.01607-25>

Re: Spectrum01607-25 (**Clinical characteristics and treatment outcomes of *Acinetobacter baumannii* bloodstream infections in a setting with high carbapenem-susceptibility among isolates**)

Dear Dr. Jinghao Nicholas Ngiam:

Thank you for the privilege of reviewing your work. Below you will find my comments, instructions from the Spectrum editorial office, and the reviewer comments.

Revision Guidelines

Sincerely,
Po-Yu Liu
Editor
Microbiology Spectrum

Reviewer #1 (Comments for the Author):

The manuscript entitled "Clinical characteristics and treatment outcomes of *Acinetobacter baumannii* bloodstream infections in a setting with high carbapenem-susceptibility among isolates" addresses a relevant topic, aiming to clarify the clinical significance and therapeutic outcomes associated with bloodstream infections caused by carbapenem-susceptible (CSAB) compared to carbapenem-resistant (CRAB) *Acinetobacter baumannii*. The study provides an interesting perspective, given that in most clinical contexts CRAB infections predominate, whereas here the authors report a relatively unusual scenario where CSAB infections are more frequent.

The methodological approach is generally clear, relying on a retrospective analysis of clinical, microbiological, and therapeutic data from a single institution. The authors clearly state their objectives and adequately employ multivariable statistical analysis to evaluate factors associated with adverse outcomes. The findings that prior carbapenem exposure, ICU-onset infections, and the definitive use of carbapenems for CSAB are independently associated with adverse outcomes, have potentially important implications for clinical practice and antimicrobial stewardship.

However, the manuscript presents several important limitations and requires substantial improvements before being suitable for publication. Specifically, the relatively small sample size (especially the CRAB group, n=15) significantly limits the generalizability of the findings. Moreover, the retrospective, observational nature of the study and the single-center setting further constrain broader applicability. The manuscript would greatly benefit from a more detailed description of patient characteristics, infection control measures, and rationale behind antibiotic choices. Additionally, a clearer discussion around potential confounders (other than the elevated C-reactive protein) influencing the decision to use carbapenems as definitive therapy in patients with more severe disease is warranted. Such adjustments would help clarify whether the reported association truly reflects causality or is confounded by indication.

In conclusion, while the manuscript addresses a clinically relevant issue and offers valuable insights, significant methodological and interpretative limitations must be carefully addressed. An expanded and nuanced discussion about clinical decision-making, infection severity adjustment, and additional data from a larger cohort or multicentric study would greatly improve the manuscript's quality and impact.

Reviewer #2 (Comments for the Author):

I read the manuscript titled "Clinical characteristics and treatment outcomes of *Acinetobacter baumannii* bloodstream infections in a setting with high carbapenem-susceptibility among isolates," SO, I have identified several comments for improvement:

Scientific and Methodological comments

1. Limitations of Retrospective Design

Issue: The retrospective nature of the study limits causal inference. It is important to explicitly acknowledge the risk of residual confounding in the discussion. Propose future prospective validation studies to strengthen the findings.

2. Small Sample Size for CRAB (n=15)

Issue: The limited number of CRAB cases may restrict the statistical power and generalizability of the results. Include this limitation in the manuscript and emphasize that the subgroup findings are exploratory in nature.

3. Limited Exploration of Microbial Genotype or Resistance Mechanisms

Issue: While CRAB resistance is mentioned, there is no genotypic characterization or molecular epidemiology included. Consider incorporating genomic data or referencing local molecular trends to enhance the analysis of resistance.

4. Potential Confounding by Indication in Antibiotic Choice

Issue: Sicker patients may have been more likely to receive carbapenems, which could confound the observed association with poor outcomes. Discuss whether C-reactive protein (CRP) adequately adjusts for disease severity, and suggest alternative severity markers or scores (e.g., SOFA, APACHE II) for future studies.

5. Lack of Data on Time to Appropriate Therapy

Issue: The timing of antibiotic administration is a critical determinant in sepsis outcomes, yet this information is not reported, this omission in the limitations and propose that future studies include this parameter.

Writing and Structure Issues

1. Minor Language Errors like:

- "A minority of were CRAB" should be corrected to "A minority were CRAB."
- "Adverse outcomes were more more likely..." contains a duplicated word. Conduct thorough proofreading to address grammar, repetitions, and clarity issues.

2. Abstract: Description of Methodology

Issue: The statistical methods (e.g., logistic regression) and variables included in the multivariable model are vaguely described. So, Specify which covariates were adjusted for in the abstract.

Data and Analysis Gaps

1. Unbalanced Use of Dual Antibiotic Therapy

Issue: The prevalence of dual therapy was significantly higher in CRAB compared to CSAB, which may confound outcome comparisons. Discuss how this difference could influence mortality or clinical improvement.

2. Lack of Antibiotic Susceptibility Profiles in CSAB Group

Issue: It is unclear how many CSAB isolates were susceptible to ceftazidime, piperacillin-tazobactam, or fluoroquinolones. Include detailed susceptibility patterns for CSAB to support conclusions regarding non-carbapenem treatment choices.

Highlight

- Unique cohort with a predominance of CSAB, contrasting with global trends.
- Clinical relevance concerning carbapenem-sparing regimens.
- Clear comparative tables that support statistical inferences.
- Expand the discussion section to address mechanisms behind higher mortality associated with carbapenem use in CSAB and broader implications for antimicrobial stewardship.
- Improve clarity and flow in the results section by enhancing transitions and avoiding redundant statistics.

Would you like me to directly revise the manuscript text to incorporate these improvements?

I read the manuscript titled "Clinical characteristics and treatment outcomes of Acinetobacter baumannii bloodstream infections in a setting with high carbapenem-susceptibility among isolates," S0, I have identified several comments for improvement:

Scientific and Methodological comments

.1 a Limitations of Retrospective Design

Issue: The retrospective nature of the study limits causal inference. It is important to explicitly acknowledge the risk of residual confounding in the discussion. Propose future prospective validation studies to strengthen the findings.

.2 Small Sample Size for CRAB (n=15)

Issue: The limited number of CRAB cases may restrict the statistical power and generalizability of the results. Include this limitation in the manuscript and emphasize that the subgroup findings are exploratory in nature.

.3 Limited Exploration of Microbial Genotype or Resistance Mechanisms

Issue: While CRAB resistance is mentioned, there is no genotypic characterization or molecular epidemiology included. Consider incorporating genomic data or referencing local molecular trends to enhance the analysis of resistance.

.4 Potential Confounding by Indication in Antibiotic Choice

Issue: Sicker patients may have been more likely to receive carbapenems, which could confound the observed association with poor outcomes. Discuss whether C-reactive protein (CRP) adequately adjusts for disease severity, and suggest alternative severity markers or scores (e.g., SOFA, APACHE II) for future studies.

.5 Lack of Data on Time to Appropriate Therapy

Issue: The timing of antibiotic administration is a critical determinant in sepsis outcomes, yet this information is not reported, this omission in the limitations and propose that future studies include this parameter.

.1 Minor Language Errors like :

- “ • A minority of were CRAB” should be corrected to “A minority were CRAB”.
- “ • Adverse outcomes were more more likely...” contains a duplicated word. Conduct thorough proofreading to address grammar, repetitions, and clarity issues.

.2 Abstract: Description of Methodology

Issue: The statistical methods (e.g., logistic regression) and variables included in the multivariable model are vaguely described. So, Specify which covariates were adjusted for in the abstract.

Data and Analysis Gaps

.1 Unbalanced Use of Dual Antibiotic Therapy

Issue: The prevalence of dual therapy was significantly higher in CRAB compared to CSAB, which may confound outcome comparisons. Discuss how this difference could influence mortality or clinical improvement.

. 2 Lack of Antibiotic Susceptibility Profiles in CSAB Group

Issue: It is unclear how many CSAB isolates were susceptible to ceftazidime, piperacillin-tazobactam, or fluoroquinolones. Include detailed susceptibility patterns for CSAB to support conclusions regarding non-carbapenem treatment choices.

Highlight

- Unique cohort with a predominance of CSAB, contrasting with global trends.
- Clinical relevance concerning carbapenem-sparing regimens.
- Clear comparative tables that support statistical inferences.
- Expand the discussion section to address mechanisms behind higher mortality associated with carbapenem use in CSAB and broader implications for antimicrobial stewardship.
- Improve clarity and flow in the results section by enhancing transitions and avoiding redundant statistics.

Would you like me to directly revise the manuscript text to incorporate these improvements?

Ref.: Ms. No. Spectrum01607-25 (Clinical characteristics and treatment outcomes of *Acinetobacter baumannii* bloodstream infections in a setting with high carbapenem-susceptibility among isolates)
Microbiology Spectrum

We thank the Editor for allowing us the opportunity to revise our manuscript and the Reviewers for the important and constructive comments. We have amended our paper in order to address the points raised by the Reviewers.

In the sections below, each of the points raised is identified and addressed with changes in the revised manuscript.

Reviewer #1 (Comments for the Author):

The manuscript entitled "Clinical characteristics and treatment outcomes of *Acinetobacter baumannii* bloodstream infections in a setting with high carbapenem-susceptibility among isolates" addresses a relevant topic, aiming to clarify the clinical significance and therapeutic outcomes associated with bloodstream infections caused by carbapenem-susceptible (CSAB) compared to carbapenem-resistant (CRAB) *Acinetobacter baumannii*. The study provides an interesting perspective, given that in most clinical contexts CRAB infections predominate, whereas here the authors report a relatively unusual scenario where CSAB infections are more frequent.

The methodological approach is generally clear, relying on a retrospective analysis of clinical, microbiological, and therapeutic data from a single institution. The authors clearly state their objectives and adequately employ multivariable statistical analysis to evaluate factors associated with adverse outcomes. The findings that prior carbapenem exposure, ICU-onset infections, and the definitive use of carbapenems for CSAB are independently associated with adverse outcomes, have potentially important implications for clinical practice and antimicrobial stewardship.

However, the manuscript presents several important limitations and requires substantial improvements before being suitable for publication. Specifically, the relatively small sample size (especially the CRAB group, $n=15$) significantly limits the generalizability of the findings. Moreover, the retrospective, observational nature of the study and the single-center setting further constrain broader applicability. The manuscript would greatly benefit from a more detailed description of patient characteristics, infection control measures, and rationale behind antibiotic choices. Additionally, a clearer discussion around potential confounders (other than the elevated C-reactive protein) influencing the decision to use carbapenems as definitive therapy in patients with more severe disease is warranted. Such adjustments would help clarify whether the reported association truly reflects causality or is confounded by indication.

In conclusion, while the manuscript addresses a clinically relevant issue and offers valuable insights, significant methodological and interpretative limitations must be carefully addressed. An expanded and nuanced discussion about clinical decision-making, infection severity adjustment, and additional data from a larger cohort or multicentric study would greatly improve the manuscript's quality and impact.

We thank the Reviewer for the feedback on our manuscript, and for providing us with constructive feedback.

We agree there are limitations with the sample size and an expansion of the dataset with other centers would increase the strength of the study. However, the appropriate statistical calculations were made and significant differences were seen in a number of the reported outcomes including (1) all-cause mortality differences between CRAB and CSAB, and (2) positive outcomes in patients with CSAB treated with carbapenem-sparing antibiotics.

We have also added more detail on patient characteristics and infection control measures in the Methods section, and clarified the rationale for carbapenem versus non-carbapenem use in CSAB (e.g., empirical coverage for severe presentations, local practice patterns, allergy considerations). Finally, we have expanded the Discussion to note the possibility of confounding by indication beyond CRP levels. Additionally, we have included the important point raised by the Reviewer that future studies should incorporate validated severity scores such as SOFA or APACHE II.

The changes are made in our manuscript (Page 6, 9).

Reviewer #2 (Comments for the Author):

I read the manuscript titled "Clinical characteristics and treatment outcomes of *Acinetobacter baumannii* bloodstream infections in a setting with high carbapenem-susceptibility among isolates," SO, I have identified several comments for improvement:

Scientific and Methodological comments

1. Limitations of Retrospective Design

Issue: The retrospective nature of the study limits causal inference. It is important to explicitly acknowledge the risk of residual confounding in the discussion. Propose future prospective validation studies to strengthen the findings.

We are grateful for this comment. We have added this important fact to our limitations (Page 9).

2. Small Sample Size for CRAB (n=15)

Issue: The limited number of CRAB cases may restrict the statistical power and generalizability of the results. Include this limitation in the manuscript and emphasize that the subgroup findings are exploratory in nature.

We thank the Reviewer for raising this important point. Indeed, we agree the small sample size (in particular, for CRAB) is an important limitation of our manuscript. We have highlighted this in our limitations as well (Page 9).

3. Limited Exploration of Microbial Genotype or Resistance Mechanisms

Issue: While CRAB resistance is mentioned, there is no genotypic characterization or molecular epidemiology included. Consider incorporating genomic data or referencing local molecular trends to enhance the analysis of resistance.

We thank the reviewer for this suggestion. We did not perform genotypic characterization on the isolates in this cohort. However, we have now referenced recent local molecular epidemiology data from our institution (Chew KL et al., Pathology 2025) describing the genotypic diversity of CRAB and CSAB isolates in Singapore, including the predominance of OXA-23 carbapenemase producers among CRAB. We have also noted in the Discussion that future studies incorporating genomic analyses of our cohort would enhance understanding of resistance mechanisms and transmission dynamics.

4. Potential Confounding by Indication in Antibiotic Choice

Issue: Sicker patients may have been more likely to receive carbapenems, which could confound the observed association with poor outcomes. Discuss whether C-reactive protein (CRP) adequately adjusts for disease severity, and suggest alternative severity markers or scores (e.g., SOFA, APACHE II) for future studies.

Indeed, we fully agree with this comment by the Reviewer. Our findings are at best exploratory, and patients who had been on carbapenems may have been sicker, and consequently have poorer outcomes (although we adjusted for CRP). Future prospective work should include other severity markers as well (e.g. SOFA and APACHE II) to validate our findings. We did not systematically capture these severity markers as not all the information was readily available on retrospective review of the electronic medical records. We have added these important points to our limitations (Page 9).

5. Lack of Data on Time to Appropriate Therapy

Issue: The timing of antibiotic administration is a critical determinant in sepsis outcomes, yet this information is not reported, this omission in the limitations and propose that future studies include this parameter.

We thank the Reviewer for this comment. Indeed, in patients with sepsis, time-to-appropriate antibiotics is an important factor that is associated with clinical outcomes. As we did not systematically collect this information, we have included this in our limitations as well (Page 9).

Writing and Structure Issues

1. Minor Language Errors like:

- "A minority of were CRAB" should be corrected to "A minority were CRAB."
- "Adverse outcomes were more more likely..." contains a duplicated word. Conduct thorough proofreading to address grammar, repetitions, and clarity issues.

Thank you for pointing out these errors in our text. We have rectified them accordingly, and also proof-read our manuscript for any other typographical errors.

2. Abstract: Description of Methodology

Issue: The statistical methods (e.g., logistic regression) and variables included in the multivariable model are vaguely described. So, Specify which covariates were adjusted for in the abstract.

Thank you for this helpful comment. Due to the small sample size and limited number of adverse outcome events in the CSAB group, we only adjusted for C-reactive protein (CRP) in the multivariable model to avoid model overfitting. We have now specified this in both the Abstract and Methods for clarity (Page 4, 7).

Data and Analysis Gaps

1. Unbalanced Use of Dual Antibiotic Therapy

Issue: The prevalence of dual therapy was significantly higher in CRAB compared to CSAB, which may confound outcome comparisons. Discuss how this difference could influence mortality or clinical improvement.

We agree with the reviewer that the markedly higher prevalence of dual therapy in CRAB (40.0% vs 1.8% in CSAB) could have influenced the comparisons. We have now included this in the Discussion that dual therapy is more often used for CRAB in our setting due to guideline recommendations and limited therapeutic options. For CSAB, no such treatment guidelines exist. The higher mortality in CRAB despite more frequent dual therapy likely reflects the resistance profile (limiting antibiotic choice) as well as more severe illness in CRAB patients rather than treatment strategy alone, but this difference may still be a source of residual confounding. This is added to our discussion (Page 7).

2. Lack of Antibiotic Susceptibility Profiles in CSAB Group

Issue: It is unclear how many CSAB isolates were susceptible to ceftazidime, piperacillin-tazobactam, or fluoroquinolones. Include detailed susceptibility patterns for CSAB to support conclusions regarding non-carbapenem treatment choices.

We thank the reviewer for this comment. Susceptibility data for CSAB isolates are presented in Table 1. We have now explicitly summarised these data in the Results section, noting that the majority of CSAB isolates were susceptible to ceftazidime (76.8%), piperacillin-tazobactam (82.1%), and ciprofloxacin (98.2%). We have also expanded the Discussion to highlight how these susceptibility patterns support the use of non-carbapenem agents in CSAB and align with our findings that carbapenem-sparing regimens were associated with favourable outcomes. This is added to our results (Page 8) and discussion (Page 9).

Highlight

- Unique cohort with a predominance of CSAB, contrasting with global trends.
- Clinical relevance concerning carbapenem-sparing regimens.
- Clear comparative tables that support statistical inferences.
- Expand the discussion section to address mechanisms behind higher mortality associated with carbapenem use in CSAB and broader implications for antimicrobial stewardship.
- Improve clarity and flow in the results section by enhancing transitions and avoiding redundant statistics.

Would you like me to directly revise the manuscript text to incorporate these improvements?

We are grateful for the comments provided by the Reviewer. We have enhanced our manuscript as suggested based on the above recommendations.

Overall, thank the Editor and Reviewers for the kind and helpful comments. We hope the paper is now suitable for publication in the Journal.

Best Regards,

Dr Nicholas Ngiam and Dr Matthew Koh

Re: Spectrum01607-25R1 (**Clinical characteristics and treatment outcomes of *Acinetobacter baumannii* bloodstream infections in a setting with high carbapenem-susceptibility among isolates**)

Dear Dr. Jinghao Nicholas Ngiam:

Your manuscript has been accepted, and I am forwarding it to the ASM production staff for publication. Your paper will first be checked to make sure all elements meet the technical requirements. ASM staff will contact you if anything needs to be revised before copyediting and production can begin. Otherwise, you will be notified when your proofs are ready to be viewed.

Sincerely,
Po-Yu Liu
Editor
Microbiology Spectrum

Reviewer #1 (Comments for the Author):

The revised manuscript has substantially improved compared to the initial version. The authors have carefully addressed the reviewers' comments by expanding the methodological details, clarifying the rationale for antibiotic choices, and providing a more nuanced discussion of limitations, including residual confounding, absence of severity indices, and lack of time-to-appropriate therapy data. The addition of local molecular epidemiology data and explicit susceptibility patterns for CSAB isolates further strengthen the clinical context and interpretability of the findings. While some inherent weaknesses (retrospective design, small sample size, absence of prospective severity scoring) cannot be fully overcome, these have now been transparently acknowledged. Overall, the manuscript is clearer, more balanced in interpretation, and provides a valuable contribution to the literature on *A. baumannii* bloodstream infections in an unusual epidemiological context.

Reviewer #2 (Comments for the Author):

no comments